# Digital Mergers and Acquisitions and Enterprise Innovation Quality: Analysis Based on Research and Development Investment and Overseas Subsidiaries

Helian Xu and Shiqi Deng *

School of Economics and Trade, Hunan University, Changsha 410006, China; xuhelian@163.com
* Correspondence: dengshiqi47@outlook.com

**Abstract:** Utilizing a hand-collected dataset on digital cross-border mergers and acquisitions (M&As), we conducted an exploratory study about the effect of digital overseas M&As on the innovative quality of acquiring enterprises. Based on the digital cross-border M&A behavior of Chinese listed firms from 2010 to 2022, we offer original and robust evidence that reveals that enterprises engaging in digital cross-border M&As are more likely to produce high-quality innovations and services, and this effect may be moderated by human capital. Our explorations specifically reveal that the increase in quality of innovation from digital cross-border M&As could occur through research and development (R&D) investment and overseas subsidiaries. In addition, we found that the positive effect is especially pronounced in enterprises located in the Eastern and Western regions, and it also exists among high-tech enterprises, relatively large-scale enterprises, and digital-acquiring enterprises. We conclude by discussing how important it is for M&A enterprises to use digital technology to shape innovation quality.

**Keywords:** digital cross-border M&As; mergers and acquisitions; innovation quality; digital technology; patent

## 1. Introduction

Enhancing the level of innovation has emerged as a crucial focus area for expediting the establishment of a new development paradigm characterized by dual circulation in China. According to the "World Intellectual Property Indicators" report released by the World Intellectual Property Organization (WIPO), China attained the status of the world's largest holder of valid patents in 2021. Nevertheless, the worth of innovation quality surpasses quantity in the context of high-quality economic development and social security, as mentioned by Makridis and McGuire [1]. Against this backdrop, the importance of examining the innovation quality of Chinese enterprises has become increasingly necessary.

With the progress in digitalization, multinational enterprises have significantly enhanced their performance. In 2021, the sales revenue of the top 100 global digital multinational corporations had increased by 1.58 times compared to six years earlier. Notably, the net income of these corporations grew by over 60% within a single year (UNCTAD, 2021). During the same period, the Ministry of Commerce, the Central Cyberspace Administration, and the Ministry of Industry and Information Technology of China collaboratively released the "Guidelines for Foreign Investment and Cooperation in the Digital Economy," encouraging enterprises to capitalize on opportunities in the digital infrastructure market overseas and elevate the digital management standards of foreign investment and cooperation. Digital enterprise M&As, as a burgeoning economic paradigm, leverage data resources as a pivotal element and utilize modern information networks as the primary conduit to facilitate enterprises in achieving strategic advancements and surpassing competitors in the international market. Chinese companies are actively involved in digital overseas M&As, pursuing transformative opportunities. For instance, in 2022, Dianlian Technology targeted

FTDI, a prominent British market leader in chipset manufacturing. Additionally, in early 2022, Haifule Group successfully concluded the acquisition of ThingOS, a German Internet of Things startup.

Therefore, will innovation, as a pivotal element in modernization and nation-building, be influenced by digital cross-border mergers and acquisitions? What are the potential impact mechanisms and heterogeneous effects? A comprehensive exploration of these inquiries holds theoretical and policy-related significance for the cohesive advancement of the worldwide digital industry and for formulating a novel paradigm for high-quality development in China.

Throughout the latest available literature, numerous scholars have extensively researched the factors influencing enterprise innovation. These factors encompass institutional investors [2,3], generalized trust among individuals [4], gender diversity in ownership [5], software piracy [6], the adoption of AI technology [7], technology standards [8], R&D subsidy programs [9], political connections [10], R&D tax credit schemes [11], clusters [12], product market competition [13], patent publications [14], labor scarcity [15], labor regulations [16], liability costs [17], and corporate venture capital investment portfolios [18].

Another avenue of innovation research delves into the influence of digitalization on enterprise technological advancements, examining this phenomenon from various perspectives. This body of literature primarily concentrates on topics such as digital transformation [19–21], the extent of digitalization [22–24], digital platforms [25], and digital technologies [26].

The most closely related literature to this study currently centers on the impact of digital cross-border M&As. Tang et al. [27] argued that compared to non-M&A digital enterprises, digital cross-border M&A enterprises exhibit a more pronounced positive market value effect. Laucis [28] conducted a study involving 20 digital cross-border M&A case studies, intending to offer relevant recommendations for decision-makers engaged in digital cross-border M&As through expert interviews and surveys. Chen et al. [29] restricted the focus to manufacturing enterprises and utilized data from 2012 to 2021 to demonstrate the positive influence of digital cross-border M&As on firms' total factor productivity (TFP). Zhou et al. [30] showed that digital cross-border M&As promoted patent applications and grants from 2006 to 2019. Hanelt et al. [31] employed the world's largest automobile manufacturing data to attest to the positive relationship between digital cross-border M&As and new digital patents.

Regrettably, the limited relevant literature currently primarily delves into the effects of cross-border M&As in the digital economy, focusing on variables such as total factor productivity (TFP) [29], digital innovation [31], and patent application and grant outcomes [30]. Moreover, there are several deficiencies: Firstly, a prior study [31] confines its examination to a specific industry as the research subject, potentially limiting the generalizability of its findings from being applied to broader industrial contexts. Secondly, antecedent research [30] lacks a delineated analysis of the mechanism of impact, introducing ambiguity regarding the precise pathways through which digital cross-border M&As influence pertinent outcomes. Thirdly, certain scholarly contributions [28] rely solely on case studies without empirical testing, potentially compromising their findings' methodological rigor and broader applicability. Fourthly, the existing literature lacks effective analytical methods and comprehensive theoretical frameworks for investigating the effects of digital cross-border M&As on the quality of innovation. This research gap assumes significance, given that innovative quality denotes fundamental inventions with heightened commercial value and technological impact [2].

To address the aforementioned gaps, this research will focus on cross-border M&As across various industries within the digital economy as the comprehensive analysis subject. Utilizing microdata from A-share listed companies in China's Shanghai and Shenzhen stock exchanges, spanning from 2010 to 2022, the study aims to empirically investigate the innovation quality effects of digital cross-border M&A activities. This research not only unveils the moderating role of human capital and elucidates the potential underlying

mechanisms through two pathways, R&D investment and the establishment of overseas subsidiaries, but also explores the heterogeneity effects from several perspectives. The findings include: (1) Digital cross-border M&As exert a driving effect on enhancing the quality of enterprise innovation; (2) the impact of digital cross-border M&As on the quality of enterprise innovation is more significant in enterprises with higher levels of human capital; (3) enterprises engaged in digital cross-border M&As enhance innovation quality by augmenting R&D investment and establishing overseas subsidiaries; (4) high-tech enterprises, relatively large-scale enterprises, those in the Eastern and Western regions, and digital-acquiring enterprises are more inclined to utilize digital cross-border M&As to enhance the quality of enterprise innovation.

The marginal contributions of this study are threefold: Firstly, aligning with the digital economy's developmental context, this research integrates digital technology with enterprise M&As, thereby extending the research boundaries of the cross-border M&A theory. Secondly, this study offers the initial empirical evidence regarding the impact of digital overseas M&As on the innovation quality of Chinese enterprises. It explores potential mechanisms from a global perspective, encompassing both domestic channels (R&D investment) and foreign channels (overseas subsidiaries), thereby enriching the research landscape in enterprise innovation and suggesting new avenues for enhancing enterprise innovation levels and promoting the high-quality development of China's economy within the digital economy framework. Thirdly, the digital industry data, selected according to the "Statistical Classification of the Digital Economy and Its Core Industries (2021)," has been meticulously organized and targeted using the CNRDS M&As database. This approach minimizes data usage limitations and provides valuable references for subsequent research. Fourthly, by adopting a human capital perspective, this study examines the potential moderating effects of digital cross-border M&As. It further investigates the heterogeneity effects of digital cross-border M&As on the quality of enterprise innovation, considering technological level, enterprise size, geographical location, and digital nature. These insights offer crucial references for enhancing the digital factor market and facilitating the development of the digital economy.

## 2. Theoretical Analysis and Hypothesis

### 2.1. Digital Cross-Border M&As and Innovative Quality

The recombination of knowledge ensures the continuity of knowledge creation [32]. Therefore, acquiring enterprises can provide direct access to digital technology through digital cross-border M&As, expanding their digital knowledge base and database by entering the knowledge networks of targeted enterprises [31,33,34], which facilitates the effective integration of internal resources [31,35,36], enhancing innovation efficiency and quality. Moreover, the acquiring enterprises can conveniently process and manage data through digital technology. The "self-reference" nature of digital technology [37] means the digital tools necessary for innovation are more affordable to a broad spectrum of previously excluded economic and innovative activities, which allows the acquiring enterprises to reduce costs related to data mining, fusion, and analysis, obtaining decision support through digital technology [38]. This results in more efficient and cost-saving data utilization to expand digital businesses and build digital capabilities [39], fostering enterprises' digital resilience, breaking through existing business areas, and prompting the probability of high-quality R&D that improves innovation quality in highly uncertain business environments.

Furthermore, the acquiring enterprises can establish effective communication bridges with users through digital platforms, accumulating more customer resources by tracking customer preferences and feedback. For instance, digital platforms can integrate and interact online and offline, reaching a wider audience through digital channels [40]. This is beneficial for the acquiring enterprises as it provides a clearer understanding of current market trends and personalized consumer needs. Acquiring enterprises gain a substantial competitive advantage through unique customer resources, enabling quick and effective responses to market changes and adjustments to products and services [41] and driving con-

tinuous innovation and development through digital empowerment. Hence, we propose the following hypothesis:

**H1:** *Digital cross-border M&As are beneficial for improving the innovative quality of acquiring enterprises.*

### 2.2. Moderating Effect: Human Capital

The process of enhancing innovative quality through digital cross-border M&As may be influenced by human capital. Firstly, the acquiring enterprises can directly augment their talent pool with individuals possessing digital thinking and skills via digital cross-border M&As, which facilitates a rapid adaptation to digitization and promotes the seamless integration of new technologies with existing production processes. The rich knowledge backgrounds and diverse perspectives on problems from globally flowing human capital contribute significantly to expanding enterprises' understanding of related issues. This, in turn, enables effective integration of national and corporate cultures [42], fostering innovative thinking in digital cross-border M&As [43]. Subsequently, this leads to a continuous dissemination of technological innovation and quality improvement [44]. Secondly, advanced human resources exhibit a heightened ability to comprehend, learn, and transform external knowledge. Effective communication and interaction among personnel can reduce resource acquisition and technological learning costs and mitigate information asymmetry [45]. This strengthens coordination and communication between enterprises, maximizing the value of various resources and supporting targeted technological innovation. Thirdly, the influx of skilled human capital into the host country stimulates competition and collaboration among individuals, resulting in an innovation network. This "peer effect" encourages highly qualified personnel to continuously capitalize on their comprehensive advantages, enhance their professional abilities, and assist enterprises in leveraging digital technology to revolutionize products, services, and business models, improving enterprise innovation quality [46]. Therefore, we raise the hypothesis as follows:

**H2:** *Human capital positively moderates the impact of digital enterprise M&As on enterprise innovation quality.*

### 2.3. Internal Channel: R&D Investment

On one side, from the perspective of R&D needs, digital cross-border M&As can leverage the unique advantages of digital technology to swiftly and accurately predict and analyze users' dynamic needs in real time [30]. This enables breaking through the information cocoon, facilitating the alignment of existing products and services, and formulating new product standards based on the preferences of advanced technology consumers. This, therefore, necessitates enterprises to invest in more innovative resources. Consequently, to meet R&D needs, enterprises escalate R&D investments to propel creative development. On the other side, concerning R&D costs, digital technology possesses the characteristic of "homogenization of data" [37], which means any digital content (audio, video, text, or image) can be stored, transmitted, processed, and displayed using the same digital devices and networks. It can encode any digital content into binary digits 0 or 1, homogenizing them (such as in storage and transportation), which enhances the ability of enterprises to search and obtain information, reduces the sunk costs of R&D investment, and incentivizes enterprises to channel more R&D resources and funds into a R&D team. This, in turn, achieves the accumulation of advanced technology and intellectual property, ultimately enhancing the quality of enterprise innovation.

Accordingly, this study presents the hypothesis:

**H3:** *Digital cross-border M&As have improved the quality of enterprise innovation through R&D investment.*

*2.4. External Channel: Overseas Subsidiaries*

The advancement of digital technology has significantly enhanced the accessibility and efficiency of information networks. The acquiring enterprises and their overseas subsidiaries establish connections and communication through digital platforms to assist the parent company in integrating and managing global resources [47], which facilitates the easier acquisition of market data, competitive intelligence, and the essential business information needed for investments in the country where the subsidiary is located. It aids companies in understanding the market environment, deciphering competitors' strategies, improving the success rate of digital cross-border M&As, and formulating market innovation strategies. Through overseas subsidiaries, external knowledge can be integrated for internal use. The challenges of obtaining information during enterprise M&As in different regions can be mitigated, which reduces the agency cost in innovation decision-making [48], thus motivating senior management to unleash entrepreneurial talent, optimizing innovation decision-making, and curbing low-quality innovation behavior within enterprises. Additionally, possessing multiple overseas subsidiaries diversifies risks. In the event of innovation failure in one market, other subsidiaries can enhance innovation quality by learning from experiences and lessons, thereby achieving a leap in R&D capabilities and innovation catch-up.

Therefore, this study proposes the hypothesis:

**H4:** *Digital cross-border M&As incentivize improving enterprise innovation quality through overseas subsidiaries.*

### 3. Data and Identification Strategy

*3.1. Sample and Data Resources*

This study utilizes a sample of Chinese A-share listed companies in the Shanghai Stock Exchange and Shenzhen Stock Exchange from 2010 to 2020. The data are processed as follows: (1) Exclude financial companies; (2) exclude companies with a listing status of "ST", "* ST", or "PT"; and (3) exclude samples with severely missing key variables. The final dataset includes 18,024 annual observations of enterprises.

The data-matching process is as follows: First, from the "Statistical Classification of Digital Economy and Its Core Industries (2021)", the 01–04 major categories are selected corresponding to the core industries of the digital economy, namely the digital industrialization part, which is the foundation of the development of the digital economy, including computer communication and other electronic equipment manufacturing industries, telecommunications, broadcasting, television and satellite transmission services, internet and related services, software, and information technology services, etc. This is matched with the CNRDS M&A database via "industry" to ultimately obtain digital cross-border M&A data. Second, the digital cross-border M&A data are matched with the patent database through the "enterprise securities code"; the patent data are sourced from the CNRDS Innovation Patent Research database. The financial data are sourced from the CSMAR database. Specifically:

1.  The first dataset comprises core industries of the digital economy data from the National Bureau of Statistics, categorizing the digital economy industry into five categories: 01, digital product manufacturing industry; 02, digital product service industry; 03, digital technology application industry; 04, digital factor-driven industry; and 05, digital efficiency improvement industry. The 01–04 categories in this classification are considered the core industries of the digital economy.
2.  The second data source is the cross-border M&A data from the CNRDS cross-border M&A database. This database includes information on targeted parties, merger events, and listed companies' acquiring parties. It contains details, such as the ID of the merger event, the effective date of the merger event, the name of the acquiring party, the name of the targeted party, the industry of the acquiring party, the industry of the targeted party, and the stock code of the acquiring party, among other information.

3. To measure enterprise financial information in our sample, we collected data on the enterprise size, total assets, net profit margin, Tobin Q, enterprise sales expense ratio, enterprise age, property nature, and all invention and utility model patents from the CSMAR database.

4. The last data source is the cited data of invention patents from the CNRDS Innovation Patent Research database. This database provides details such as the stock code of the listed company, cited patent number, cited year, company type, invention type, and the number of citations in each year, excluding self-citations.

### 3.2. Variable Construction

#### 3.2.1. Dependent Variable

Innovation: This study refers to the research of Moser et al. [49] and Mao & Zhang [50], using the number of cited patents for invention applications as a proxy variable for the quality of enterprise innovation.

#### 3.2.2. Independent Variable

DMA: Using digital cross-border merge and acquisition (DMA) as a dummy variable, if a Chinese company acquires a foreign company in the digital industries (the corresponding 01–04 categories in the Statistical Classification of Digital Economy and Its Core Industries (2021) belong to the core industries of the digital economy), the DMA is set to 1, otherwise 0.

#### 3.2.3. Control Variables

Referring to the recent literature [46,51], we selected the following control variables. To alleviate the endogeneity issues, we uniformly treated the following variables with a lag of one period: The enterprise size (Size), represented by the logarithm of total assets; the net profit margin on total assets (ROA), calculated as the net profit/average balance of the total assets; the Tobin Q value (Tobin_Q), calculated as (value of circulating stock market + the number of non-circulating shares * net assets per share + book value of liabilities)/total assets; the enterprise sales expense rate (Selexprt), calculated as sales expenses/operating income; the firm's age (Firm_age), calculated as the logarithm of (the year of establishment + 1); and Ownership (SOE), 1 being for the state-owned enterprises or otherwise 0. The descriptive statistics are shown in Table 1.

**Table 1.** Descriptive statistics.

| Variable | Obs | Mean | Std.Dev. | Min | Max |
|---|---|---|---|---|---|
| Innovation | 24,417 | 3.058 | 1.782 | 0 | 8.082 |
| DMA | 38,580 | 0.002 | 0.046 | 0 | 1 |
| ROA | 28,255 | 0.040 | 0.066 | −0.232 | 0.222 |
| Tobin_Q | 29,566 | 2.067 | 1.317 | 0.857 | 8.587 |
| Size | 30,035 | 3.093 | 0.055 | 2.990 | 3.255 |
| Selexprt | 29,531 | 7.507 | 9.021 | 0.080 | 48.43 |
| Firm_age | 30,035 | 2.884 | 0.357 | 0.693 | 4.174 |
| SOE | 37,647 | 0.344 | 0.475 | 0 | 1 |

### 3.3. Empirical Model

This study endeavors to assess the influence of digital cross-border M&As on the innovative quality of the acquiring enterprises. Given the dependent variable represents count data and exhibits overdispersion, employing a panel-negative binomial regression model is deemed more suitable. The model is formulated as follows:

$$Innovation_{it} = \beta_0 + \beta_1 * DMA_{it} + \beta_2 * Control_{it-1} + \varepsilon_i + \alpha_t + \theta_{it}, \tag{1}$$

where $Innovation_{it}$ represents the innovative quality of the acquiring enterprises, which is proxied by the number of cited patents for the invention applications produced by enterprise $i$ in year $t$; $DMA_{it}$ is a dummy variable that equates to 1 if the enterprise is engaged in digital cross-border M&As, and 0 otherwise. $Control_{it-1}$ represents the firm-level control variable. In addition, this study also controls for individual fixed-effects $\alpha_t$ and time-fixed-effects $\varepsilon_i$, and $\theta_{it}$ is a random error term.

## 4. Empirical Results

### 4.1. Baseline Regression

In the empirical assessment of Equation (1) using a negative binomial regression model, the benchmark regression results are presented in column (1) of Table 2. The findings indicate a significant positive impact of digital cross-border M&As on the innovation quality of enterprises, even after accounting for all the control variables. Specifically, for each instance of engagement in digital cross-border M&As, the enterprise's innovation quality shows a noteworthy increase of 0.354 units. In summary, the benchmark regression results substantiate Hypothesis 1 of this study. This hypothesis posits that companies involved in digital cross-border M&As enhance their digital business and cultivate digital capabilities by directly accessing technology, utilizing data at a low cost, and exploring potential customers through digital platforms. Consequently, these companies continuously elevate their innovation quality through digital empowerment.

**Table 2.** Baseline results and robustness checks.

| Variable | Baseline Results | Robustness Checks | | | | | | |
|---|---|---|---|---|---|---|---|---|
| | (1) | (2) | (3) | (4) | (5) | (6) | (7) | (8) |
| DMA | 0.354 *** | 0.273 ** | 0.102 *** | 0.316 ** | 0.282 ** | 0.533 *** | 0.469 *** | 0.548 *** |
| | (0.009) | (0.030) | (0.000) | (0.025) | (0.044) | (0.001) | (0.001) | (3.86) |
| ROA | −0.990 *** | −0.762 *** | −0.386 *** | −0.873 *** | −0.890 *** | −0.734 *** | −0.926 *** | −0.941 *** |
| | (0.000) | (0.000) | (0.000) | (0.000) | (0.000) | (0.000) | (0.000) | (−8.40) |
| Tobin_Q | 0.020 *** | 0.017 *** | 0.017 *** | 0.024 *** | 0.025 *** | 0.022 *** | 0.019 *** | 0.0215 *** |
| | (0.000) | (0.000) | (0.000) | (0.000) | (0.000) | (0.006) | (0.000) | (5.54) |
| Size | 1.437 *** | 0.852 *** | 1.620 *** | 1.438 *** | 1.499 *** | 1.698 *** | 1.401 *** | 1.771 *** |
| | (0.000) | (0.000) | (0.000) | (0.000) | (0.000) | (0.000) | (0.000) | (10.85) |
| Selexprt | 0.001 | 0.001 | 0.007 *** | 0.000 | 0.001 | 0.001 | 0.001 | 0.000290 |
| | (0.371) | (0.193) | (0.000) | (0.658) | (0.540) | (0.545) | (0.434) | (0.33) |
| Firm_age | 0.169 *** | 0.091 *** | 3.033 *** | 0.127 *** | 0.140 *** | 0.104 *** | 0.146 *** | 0.167 *** |
| | (0.000) | (0.000) | (0.000) | (0.000) | (0.000) | (0.007) | (0.000) | (5.92) |
| SOE | 0.234 *** | 0.161 *** | 0.256 *** | 0.236 *** | 0.271 *** | 0.245 *** | 0.237 *** | 0.300 *** |
| | (0.000) | (0.000) | (0.000) | (0.000) | (0.000) | (0.000) | (0.000) | (15.96) |
| pro_GDP | | | | | 0.000 *** | | | |
| | | | | | (0.000) | | | |
| Structure | | | | | −0.624 *** | | | |
| | | | | | (0.000) | | | |
| _cons | −5.834 *** | −3.649 *** | −17.492 *** | −5.745 *** | −5.776 *** | −6.557 *** | −5.690 *** | −6.942 *** |
| | (0.000) | (0.000) | (0.000) | (0.000) | (0.000) | (0.000) | (0.000) | (−13.75) |
| Year fixed effects | Yes | Yes | Yes | Yes | Yes | Yes | Yes | Yes |
| Firm fixed effects | Yes | Yes | Yes | Yes | Yes | Yes | Yes | Yes |
| N | 18,024 | 18,040 | 18,024 | 11,730 | 11,730 | 7182 | 16,838 | 14,225 |

$p$-values in parentheses with * $p < 0.1$, ** $p < 0.05$, and *** $p < 0.01$. This table combines the baseline results and robustness checks, with column (1) being the baseline regression test and columns (2)–(8) being the robustness tests. This study uses robust standard errors, the same as below.

### 4.2. Robustness Checks

In this section, we proceed with the additional robustness checks to validate the findings from the baseline regression.

#### 4.2.1. Replacing the Dependent Variable

This study explores an alternative measure for innovation quality, utilizing the number of cited invention patents. The results in column (2) of Table 2 demonstrate a consistently

positive and statistically significant relationship between digital cross-border M&As and corporate innovation performance. These findings affirm the robustness of the regression results, reinforcing the conclusion that engagement in digital cross-border M&As is associated with a positive impact on innovation performance.

### 4.2.2. Replacing Estimation Methodology

This study assesses the robustness of the main regression by employing the Poisson regression method as an alternative to the panel-negative binomial regression. The results in column (3) of Table 2 consistently indicate significant positive effects of digital cross-border M&As on innovation. This reaffirms the robustness of the main results, supporting the conclusion that engaging in digital cross-border M&As is associated with a positive impact on innovation performance.

### 4.2.3. Excluding Patent Citation Data That Have Been Cited for Less Than Three Years

To mitigate the value bias associated with patent citation data, only the patent information published within the previous three years cited in the company's patent application is counted. This study excludes data collected during the period 2020 to 2022, The regression results, presented in column (4) of Table 2, demonstrate that the estimated coefficients of the key variables align with those in column (1) of Table 2. This consistency further supports the robustness of the findings, reinforcing the conclusion that the positive effects of digital cross-border M&As on innovation persist, even when excluding data collected during the period 2020 to 2022.

### 4.2.4. Adding Macro Variables

With reference to Zhao and Shi [52], to account for macro-level impacts, this study conducts a robustness analysis by incorporating the provincial GDP and provincial secondary industry structure (secondary industry output value/total output value) into the regression, as reported in column (5) of Table 2. The results consistently align with the basic regression, indicating that the positive effects of digital cross-border M&As on innovation remain robust, even after controlling for these additional macroeconomic factors.

### 4.2.5. Removing Enterprises in Provincial Capital Cities and Municipalities

To mitigate the potential impact of developed economies and ample resource endowments in provincial capital cities and municipalities directly under the central government, this study excludes data from these regions in the regression analysis. The results, presented in column (6) of Table 2, indicate that the study's main findings remain robust, even after this exclusion, which suggests that the positive effects of digital cross-border M&As on innovation persist when accounting for the potential influence of these specific regions.

### 4.2.6. Avoiding Result Bias Caused by Unpatentable Data

Due to the fact that data, software, and machine learning models are not patentable, choosing patents as a measure of innovation quality across the entire sample may lead to biased results. Therefore, we excluded data from non-manufacturing enterprises and removed the "Software and Information Technology Services Industry" with code classification "I65" from the "China's Industrial classification for national economic activities" to address the problem of sample selection bias caused by non-patchable data, software, and machine learning models. The results, presented in columns (7) and (8) of Table 2, indicate that, after excluding non-patentable data, the baseline regression remains robust.

## 5. Further Analysis

### 5.1. Moderating Effects

To assess the moderating effect of human capital, this study utilizes the total provincial human generalized capital per capita, as published by the Center for Human Capital and Labor Economics at the Central University of Finance and Economics, as the moderating

variable. An interaction term, represented by digital * HR, is added to the benchmark model. The results in column (1) of Table 3 demonstrate that the interaction term is significantly positive, which suggests that human capital positively moderates the relationship between digital cross-border M&As and enterprise innovation. Specifically, higher human capital strengthens the innovative quality effect of digital cross-border M&As, confirming Hypothesis 2.

**Table 3.** Moderating effects and mechanism analysis.

| Variable | Innovation | Innovation | RDexp | Innovation | Innovation | Subsidiaries | Innovation |
|---|---|---|---|---|---|---|---|
| | (1) | (2) | (3) | (4) | (5) | (6) | (7) |
| DMA | 1.264 *** | 0.354 *** | 0.380 *** | 0.373 *** | 0.354 *** | 0.486 *** | 0.295 ** |
| | (0.001) | (0.009) | 0.000 | (0.006) | (0.009) | (0.000) | (0.037) |
| ROA | −0.899 *** | −0.990 *** | −0.172 * | −1.026 *** | −0.990 *** | −0.284 ** | −0.879 *** |
| | (0.000) | (0.000) | (0.000) | (0.000) | (0.000) | (0.025) | (0.000) |
| Tobin_Q | 0.023 *** | 0.020 *** | 0.001 ** | 0.018 *** | 0.020 *** | 0.016 *** | 0.014 *** |
| | (0.000) | (0.000) | (0.024) | (0.000) | (0.000) | (0.000) | (0.009) |
| Size | 1.433 *** | 1.437 *** | 0.501 | 1.521 *** | 1.437 *** | 1.133 *** | 0.809 *** |
| | (0.000) | (0.000) | (0.000) | (0.000) | (0.000) | (0.000) | (0.000) |
| Selexprt | 0.000 | 0.001 | 0.000 | 0.001 | 0.001 | 0.001 | −0.001 |
| | (0.713) | (0.371) | (0.765) | (0.493) | (0.371) | (0.166) | (0.381) |
| Firm_age | 0.130 *** | 0.169 *** | −0.008 | 0.191 *** | 0.169 *** | 0.125 *** | 0.196 *** |
| | (0.000) | (0.000) | (0.706) | (0.000) | (0.000) | (0.000) | (0.000) |
| SOE | 0.250 *** | 0.234 *** | 0.083 *** | 0.231 *** | 0.234 *** | −0.041 * | 0.361 *** |
| | (0.000) | (0.000) | (0.000) | (0.000) | (0.000) | (0.055) | (0.000) |
| DMA*HR | 0.225 ** | | | | | | |
| | (0.015) | | | | | | |
| HR | 0.031 *** | | | | | | |
| | (0.000) | | | | | | |
| RDexp | | | | 0.000 *** | | | |
| | | | | (0.000) | | | |
| Subsidiaries | | | | | | | 0.012 *** |
| | | | | | | | (0.000) |
| _cons | −5.851 *** | −5.834 *** | −2.289 *** | −6.111 *** | −5.834 *** | −1.550 * | −3.116 *** |
| | (0.000) | (0.000) | (0.000) | (0.000) | (0.000) | (0.064) | (0.000) |
| Year fixed effects | Yes | Yes | Yes | Yes | Yes | Yes | Yes |
| Firm fixed effects | Yes | Yes | Yes | Yes | Yes | Yes | Yes |
| N | 11,508 | 18,024 | 20,146 | 15,154 | 18,024 | 10,177 | 7601 |

$p$-values in parentheses with * $p < 0.1$, ** $p < 0.05$, and *** $p < 0.01$. R&D investment data comes from the CNRDS database, overseas subsidiaries data comes from the CSMAR database, and the author manually compiled the table.

*5.2. Mechanism Analysis*

5.2.1. Enterprise R&D Investment

On the one hand, to meet the dynamic needs of enterprises in tracking and predicting users and achieving precise marketing for target customer groups, digital cross-border M&A companies will increase R&D investment, thereby optimizing the existing products, continuously increasing product production and thus enhancing the technical content of products, which is consistent with the study by Chen et al. [29]. On the other hand, enterprises also improve their operational efficiency and reduce operating costs through digital cross-border M&As, thereby investing more R&D funds and resources into product innovation to accelerate the pace of innovation and R&D. Therefore, this study uses enterprise R&D investment as a mediator variable. The regression results are shown in columns (2)–(4) of Table 3 and demonstrate that digital cross-border M&As, indeed, foster innovation quality by boosting R&D investment, confirming the validation of Hypothesis 3.

5.2.2. Overseas Subsidiaries

The presence of overseas subsidiaries proves instrumental for digital cross-border M&A companies in mitigating the agency costs associated with innovation decision-making. Through mechanisms such as knowledge sharing, technology integration, and the alleviation of information acquisition challenges inherent in remote M&As, overseas

subsidiaries contribute to a reduction in agency costs [48]. Consequently, this facilitates senior management in harnessing entrepreneurial talent for differentiated innovation, thereby enhancing overall innovation performance within the organization. This study introduces the number of overseas subsidiaries as a mediator variable to assess this intricate dynamic. The regression results, presented in columns (5)–(7) of Table 3, provide robust evidence that digital cross-border M&As, indeed, foster corporate innovation through the establishment of overseas subsidiaries. This substantiates Hypothesis 4, confirming that the number of overseas subsidiaries mediates the positive impact of digital cross-border M&As on innovation.

*5.3. Heterogeneous Tests*

5.3.1. Industry Heterogeneity

Following the OECD regulations, this study categorizes computer-related, electronics, information technology, biopharmaceutical, and communication industries as high-tech industries. By contrast, the remaining industries are classified as non-high-tech industries. In columns (1) and (2) of Table 4, a comparison of the two subsamples reveals that digital overseas M&As, undertaken by enterprises within the high-tech industry, significantly promote an improvement in enterprise innovation. At the same time, such effects are not significant for non-high-tech enterprises. A plausible explanation lies in the fact that non-high-tech enterprises encompass a substantial number of traditional industries, such as wood, coal, and transportation, characterized by relatively modest R&D investment and, consequently, insufficient motivation for innovation. Moreover, non-high-tech enterprises pursue digital overseas M&As primarily for competitive advantages, whereas high-tech enterprises are often driven by advanced technology and R&D resources.

**Table 4.** Heterogeneity tests.

| Variable | High-Tech | Non-High-Tech | Big Scale | Small Scale | Easter | Central | Western | Digital | Non-Digital |
|---|---|---|---|---|---|---|---|---|---|
| | (1) | (2) | (3) | (4) | (5) | (6) | (7) | (8) | (9) |
| DMA | 0.384 *** | −0.323 | 0.324 * | 0.162 | 0.272 * | 0.507 | 0.946 * | 0.339 * | 0.071 |
| | (0.006) | (0.491) | (0.060) | (0.462) | (0.067) | (0.239) | (0.050) | (0.082) | (0.714) |
| ROA | −1.166 *** | −0.792 *** | −0.682 *** | −1.209 *** | −1.062 *** | −0.874 *** | −0.488 | −1.237 *** | −1.000 *** |
| | (0.000) | (0.000) | (0.000) | (0.000) | (0.000) | (0.000) | (0.169) | (0.000) | (0.000) |
| Tobin_Q | 0.024 *** | 0.018 ** | 0.010 * | 0.029 *** | 0.022 *** | 0.004 | 0.026 *** | 0.010 | 0.023 *** |
| | (0.000) | (0.014) | (0.057) | (0.000) | (0.000) | (0.656) | (0.000) | (0.434) | (0.000) |
| Size | 2.241 *** | 1.020 *** | −0.265 | 2.213 *** | 1.829 *** | 0.735 ** | 0.536 | 1.817 *** | 1.552 *** |
| | (0.000) | (0.000) | (0.225) | (0.000) | (0.000) | (0.035) | (0.197) | (0.000) | (0.000) |
| Selexprt | 0.000 | −0.001 | −0.000 | 0.001 | 0.001 | 0.002 | 0.001 | −0.007 *** | 0.002 ** |
| | (0.770) | (0.284) | (0.848) | (0.346) | (0.572) | (0.256) | (0.756) | (0.003) | (0.023) |
| Firm_age | 0.192 *** | 0.212 *** | 0.144 *** | 0.117 *** | 0.218 *** | −0.077 | 0.229 *** | 0.198 *** | 0.187 *** |
| | (0.000) | (0.000) | (0.000) | (0.000) | (0.000) | (0.259) | (0.003) | (0.006) | (0.000) |
| SOE | 0.280 *** | 0.374 *** | 0.161 *** | 0.187 *** | 0.247 *** | 0.378 *** | 0.241 *** | 0.397 *** | 0.220 *** |
| | (0.000) | (0.000) | (0.000) | (0.000) | (0.000) | (0.000) | (0.000) | (0.000) | (0.000) |
| _cons | −8.439 *** | 30.319 *** | −0.528 | −7.928 *** | −7.081 *** | −3.159 *** | −3.546 *** | −7.296 *** | −6.042 *** |
| | (0.000) | (0.000) | (0.444) | (0.000) | (0.000) | (0.003) | (0.005) | (0.000) | (0.000) |
| Year fixed effects | Yes | Yes | Yes | Yes | Yes | Yes | Yes | Yes | Yes |
| Firm fixed effects | Yes | Yes | Yes | Yes | Yes | Yes | Yes | Yes | Yes |
| N | 13,173 | 4851 | 7716 | 10,308 | 12,999 | 2970 | 2055 | 2295 | 15,729 |

$p$-values in parentheses with * $p < 0.1$, ** $p < 0.05$, and *** $p < 0.01$. The classification standards for heterogeneity testing are derived from the classification standards of the CSMAR database, and the author manually compiled this table.

5.3.2. Heterogeneity of Enterprise Scale

This study segregates enterprises into relatively large and small categories based on whether their size exceeds the industry average to explore the diverse impacts of digital cross-border M&As on innovative quality across enterprises of varying sizes. The regression outcomes presented in columns (3) and (4) of Table 4 indicate that digital cross-border M&As by relatively large enterprises prove advantageous for enhancing their innovation level. At the same time, no significant impact is observed for relatively small enterprises. This distinction may be attributed to the enhanced capabilities of relatively large-scale enterprises to acquire digital technology, exhibit more astute market capture abilities, and allocate greater funds and resources to digital cross-border M&A activities.

These advantages empower larger enterprises to benefit more substantially in improving innovation performance through digital overseas M&As.

### 5.3.3. Regional Heterogeneity

This study dissects the sample into Eastern, Central, and Western regions, with the regression results outlined in columns (5)–(7) of Table 4. The discerned innovation quality effect of digital overseas M&As on enterprises in the Eastern and Western regions proves significantly positive, whereas insignificance is observed in the Central region. Advanced marketization in the Eastern region, coupled with its abundance of digital infrastructure and developed human capital, positions it with superior capabilities for both the absorption and application of digitization. Conversely, despite a relatively lower economic standing, the Western region is undergoing swift digital development, presenting substantial room for progress and heightened marginal benefits of innovation through digital cross-border M&As. However, the sluggish pace of digital economy growth in the Central region results in a pronounced "Central Collapse," exacerbating issues. Digital cross-border M&As do not foster a conducive environment for stability or sustained innovation and growth in the Central region. The "East-to-West Computing" project in China further facilitated complementary collaboration by aligning computing power demand in the Eastern region with land, energy, and other resources in the Western region. This initiative has contributed, to some extent, to the stimulation of digital and innovation development among enterprises in the Western region.

### 5.3.4. Digital Nature Heterogeneity

This study segregates the sample into digital- and non-digital-acquiring enterprises. Companies falling under industries such as computer, communication, and other electronic equipment manufacturing, as well as internet and related services, telecommunications, broadcasting and satellite transmission services, and software and information technology services, are classified as digital enterprises. The regression results are depicted in columns (8) and (9) of Table 4. The impact on innovation quality for digital-acquiring enterprises is significantly positive, while their non-digital counterparts are insignificant. The plausible explanation is that digital enterprises possess a more substantial knowledge stock, a narrower digital technology gap, and enhanced capabilities in digital technology integration compared to non-digital enterprises. Consequently, digital-acquiring enterprises experience a notable improvement in innovation quality.

## 6. Conclusions and Implications

### 6.1. Conclusions

In this study, we explored the relationship between digital cross-border M&As and the innovation quality of acquiring enterprises in China, utilizing a unique sample of Chinese listed firms from 2010 to 2020. Our findings indicate that (1) digital cross-border M&As positively drive an improvement in enterprise innovation quality; (2) the impact of digital cross-border M&As on innovation quality is more pronounced in enterprises with higher levels of human capital; (3) enterprises engaging in digital cross-border M&As enhance innovation quality through increased R&D investment and overseas subsidiaries; and (4) the influence of digital cross-border M&As on innovation quality is more significant for firms located in the eastern and western regions, as well as for high-tech enterprises, digital enterprises, and relatively large-scale enterprises.

### 6.2. Implications

Our study holds important policy implications. For enterprises, it is crucial to capitalize on the opportunities and advantages of the digital economy. Engaging in digital cross-border M&As allows firms to acquire, integrate, and apply cutting-edge digital technologies and resources, fostering innovation within domestic enterprises. Enterprises should first accelerate their digitalization process, transform the entire business process

through digital technology, comprehensively achieve digital reform from R&D and production to management, and provide a guarantee of solid efficiency for promoting the impact of digital mergers and acquisitions on enterprise innovation. Simultaneously, companies should establish robust knowledge and resource bases to provide reliable technical support and mitigate potential risks in future digital transformations. Additionally, enhancing digital expertise and proficiency, recruiting digital technology specialists, and accelerating the spillover effects of digital technology can further stimulate the innovation capabilities of human capital.

Digital cross-border M&A enterprises should adopt a dual approach. The accumulation of digital technology obtained through digital mergers and acquisitions drives the establishment of a digital technology knowledge base for enterprises, maximizing the value of digital mergers and acquisitions. Enterprises must also increase their R&D investment and maximize the utilization of overseas subsidiaries to enhance innovation quality. Developing strategic R&D plans, transforming and applying R&D achievements, and prioritizing intellectual property protection are essential for sustained innovation. Building a network of overseas subsidiaries enables digital cross-border M&A enterprises to obtain external information for innovation activities, continually elevating their innovation levels.

Government attention should be directed toward the development of core economic industries and the equitable allocation of resources. Providing robust digital infrastructure and hardware equipment for small- and medium-sized enterprises, particularly those in the Central region, can facilitate digital transformation. Governments should also promote the digital transformation of non-technical and traditional industries, breaking down technological innovation barriers through digital technology empowerment.

Addressing the challenges encountered by enterprises during the digital cross-border M&A process requires formulating laws and regulations tailored to the characteristics of the digital economy era. Policies related to data security, privacy protection, innovation and entrepreneurship support, market opening, financial aid, education and talent development, and intellectual property protection should be established to create a conducive environment for digital investment and development. This comprehensive approach ensures the acquisition of advanced overseas digital technology to support domestic enterprises while fully mobilizing the vitality of domestic digital innovation.

### 6.3. Limitations

First, this article only studied the innovation effects of the acquiring enterprises. In fact, during the process of digital cross-border M&As, the innovation effects of the acquired enterprises will also be affected. However, this section has not been fully discussed due to data limitations. Second, data security risks have hampered the digitization of various entities. Countries worldwide have implemented data protection measures and issued regulations and laws, such as the European General Data Protection Regulation (GDPR) and the US Data Privacy and Protection Act. In digital cross-border enterprises, data protection measures from other countries may hinder digital M&As and their innovative effects. Data protection measures and how they affect the relationship between digital cross-border M&As and innovation is another important direction that this study can expand on.

**Author Contributions:** Conceptualization, H.X. and S.D.; methodology, S.D.; software, S.D.; validation, H.X. and S.D.; formal analysis, S.D.; investigation, S.D.; resources, H.X.; data curation, H.X.; writing—original draft preparation, S.D.; writing—review and editing, H.X. and S.D.; visualization, S.D.; supervision, H.X.; project administration, H.X. All authors have read and agreed to the published version of the manuscript.

**Funding:** This research received no external funding.

**Institutional Review Board Statement:** Not applicable.

**Informed Consent Statement:** Not applicable.

**Data Availability Statement:** The data presented in this study are available on request from the corresponding author. The data are not publicly available due to restrictions, e.g., privacy or ethical reasons.

**Conflicts of Interest:** The authors declare no conflicts of interest.

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
