# Peer review of "Digital Mergers and Acquisitions and Enterprise Innovation Quality: Analysis Based on Research and Development Investment and Overseas Subsidiaries"

_sustainability, doi:10.3390/su16031120_

Round 1

Reviewer 1 Report

Comments and Suggestions for Authors

line 398, 352- resources must be added.

Table 2 needs to be explained - robustess ? line 297

What are the limits of research? The regresion in this form isnt sufficient describe.

Author Response

Dear Reviewer,

Thank you very much for your comments and professional advice. These opinions help to improve the academic rigor of our article. Based on your suggestions and request, we have corrected the revised manuscript's modifications. We hope that our work can be improved again.

Please see the attachment about the specific modifications.

Reviewer 2 Report

Comments and Suggestions for Authors

Dear authors and editor,

Thank you for the opportunity to review the submitted article.

The article titled "Digital Merge & Acquisitions and Enterprise Innovation Quality: Analysis Based on Research & Development Investment and Overseas Subsidiaries " examines the impact of digital merger and acquisition (M&A) transactions abroad on the innovation quality of enterprises.
The authors focus on Chinese firms listed on the stock exchange from 2010 to 2022, analyzing how engagement in digital M&As affects the innovation capabilities of companies. The study considers various factors, such as investments in research and development and the ownership of overseas subsidiaries and examines their impact on innovation quality in the context of different regions and types of enterprises.

Following are some comments on the article, highlighting strengths and weaknesses, along with recommendations.

Introduction:

1.       The authors effectively establish the background of the study, referring to current trends and challenges in the Chinese digital economy and in the global context of innovation.

2.       A clear research gap in the literature concerning the impact of digital M&As on innovation quality has been identified, justifying the need for conducting research.

3.       The authors clearly define the aim and scope of the research, focusing on analyzing the impact of digital M&As on innovation quality in various industries and contexts.

Theoretical analysis and hypothesis:

1.       The section develops ideas presented in the introduction, thoroughly analyzing factors that can affect innovation quality in the context of digital M&As.

2.       The authors formulate four clear hypotheses that are directly related to the research objectives of the article.

3.       The section may be difficult to understand for readers unfamiliar with the subject, due to the complex language and density of information.

4.       Some parts may seem repetitive, particularly in the context of detailed descriptions of digital M&As and their impact on innovation.

Data and identification strategy:

  1. The use of the Negative Binomial Regression Model for data analysis is appropriate.
  2. There is a lack of more detailed description of the data collection process and sample selection, which is crucial for understanding how the data were obtained and whether they are representative.
  3. Although the authors mention the inclusion of control variables, detailed information about them is limited.
  4. Every statistical model has its assumptions and potential errors. The absence of discussion on this topic can limit the understanding and interpretation of the results.

Further analysis:

  1. The section clearly presents the research findings for individuals familiar with advanced statistical methods.

1.       The section contains practical recommendations for businesses and governments regarding the utilization of benefits from digital M&As, as well as guidelines for supporting innovation and digital transformation.

2.       The authors do not discuss the limitations of their study, which is important for understanding the scope and significance of the conclusions.

3.       Although the conclusions have practical implications, there is a lack of more detailed information on how businesses can implement the recommended strategies.

Although your study provides valuable insights into the impact of digital mergers and acquisitions (M&A) on the quality of innovation in enterprises, I have noticed a lack of in-depth discussion analyzing the obtained results. I would suggest adding a section that would discuss the research findings in the context of existing theories and research in this field. I have also observed that your article does not include a clear confrontation of the obtained research results with the existing subject literature. Such a confrontation is crucial for understanding how your study fits into the current state of knowledge.

Author Response

(The authors gave the same response as above.)

Reviewer 3 Report

Comments and Suggestions for Authors

This paper presents an analysis of the cross-border merge and acquisitions related to digital enterprises.  The research topic is interesting.  However, the paper suffers the following problems.

1. The definition of "digital cross-border merge and acquisition" is ill-defined.

2. The innovation quality is not formally defined or described in details in the paper. If the innovation quality is the log function of "the number of cited patents for invention applications" as listed in Table 1,  this definition may be biased and not reliable in my opinion.  Data, software and machine learning models are not patentable.

Overall, I believe the analysis contains flaws and the findings may be misleading.

Comments on the Quality of English Language

The paper is difficult to read and follow.  It is probably due to bad translations.  Moreover, there are numerous grammatical errors and typos throughout the paper.  For example, "Innovarion" in Equation (1) should be "Innovation".  I suggest the authors to hire a professional English editor to proofread the paper and improve the English of the paper.

Author Response

(The authors gave the same response as above.)

Round 2

Reviewer 3 Report

Comments and Suggestions for Authors

The authors responded my previous comments.  In my previous comments, I mentioned that "The innovation quality is not formally defined or described in details in the paper. If the innovation quality is the log function of 'the number of cited patents for invention applications' as listed in Table 1, this definition may be biased and not reliable in my opinion. Data, software and machine learning models are not patentable".  The authors responded that machine learning is patentable.  To my best knowledge, machine learning algorithms are patentable, but not machine learning models.  In addition,  as data and software are not patentable, even they are protected in other ways, they are not included in the definition of "innovation quality" described in the paper.  In my opinion, the findings of the paper can be completely misleading and biased.

Comments on the Quality of English Language

There are still grammatical errors throughout the paper.

Author Response

Dear editor,
Thank you very much for your valuable and insightful suggestions.
We have provided some explanations and modifications based on your feedback, and we have included the details in the Word document. Please see the attachment.

Best wishes,

Round 3

Reviewer 3 Report

Comments and Suggestions for Authors

The authors responded my previous comments by arguing that (1) some software are patentable and (2) previous research used the same definition.  My concerns are as follows.

(1) This paper studies digital companies which mainly use data, software and machine learning models to compete with the competitors.  If data, software and machine learning models are not included in the definition of "innovation quality", the conclusions may be completely misleading and biased.

(2) The definition of "innovation quality" used in previous research may have specific context.  For example, for many traditional companies such as manufacturing companies, it is perfectly fine to use number of patents to measure innovation quality.  I strongly oppose to use the same definition as the previous research for digital companies that rely on non-patentable innovations to compete and survive.

Comments on the Quality of English Language

No significant changes have been made with respect to the English grammar.